# ThelR547v1—An Asymmetric Dilated Convolutional Neural Network for Real-time Semantic Segmentation of Horticultural Crops

**DOI:** 10.3390/s22228807

**Published:** 2022-11-15

**Authors:** Md Parvez Islam, Kenji Hatou, Takanori Aihara, Masaki Kawahara, Soki Okamoto, Shuhei Senoo, Kirino Sumire

**Affiliations:** 1The United Graduate School of Agricultural Sciences, Ehime University, Matsuyama 790-8566, Ehime, Japan; 2Graduate School of Agriculture, Ehime University, Matsuyama 790-8566, Ehime, Japan

**Keywords:** dilated convolution, multi-branch, asymmetric, semantic segmentation

## Abstract

Robust and automated image segmentation in high-throughput image-based plant phenotyping has received considerable attention in the last decade. The possibility of this approach has not been well studied due to the time-consuming manual segmentation and lack of appropriate datasets. Segmenting images of greenhouse and open-field grown crops from the background is a challenging task linked to various factors such as complex background (presence of humans, equipment, devices, and machinery for crop management practices), environmental conditions (humidity, cloudy/sunny, fog, rain), occlusion, low-contrast and variability in crops and pose over time. This paper presents a new ubiquitous deep learning architecture ThelR547v1 (Thermal RGB 547 layers version 1) that segmented each pixel as crop or crop canopy from the background (non-crop) in real time by abstracting multi-scale contextual information with reduced memory cost. By evaluating over 37,328 augmented images (aug1: thermal RGB and RGB), our method achieves mean IoU of 0.94 and 0.87 for leaves and background and mean Bf scores of 0.93 and 0.86, respectively. ThelR547v1 has a training accuracy of 96.27%, a training loss of 0.09, a validation accuracy of 96.15%, and a validation loss of 0.10. Qualitative analysis further shows that despite the low resolution of training data, ThelR547v1 successfully distinguishes leaf/canopy pixels from complex and noisy background pixels, enabling it to be used for real-time semantic segmentation of horticultural crops.

## 1. Introduction

Traditional horticultural crop monitoring requires manual observation, which is time-consuming and labor-intensive. The measurement process is also prone to errors due to subjective perceptions of crop variety, shape, texture, color, etc. [1,2]. However, timely monitoring of the growth, water stress status, nutrient or health status of cultivated crops is necessary to increase crop yields, reduce production costs, and reduce environmental impact.

Pixel-level image classification techniques based on semantic segmentation help to obtain information and locations of object boundaries in images for precise applications [3,4]. Although multiple authors have reported multiple successes, manual image labelling for semantic segmentation is a challenging task in agriculture, and the core constraints can be listed as follows. (1) Crop varieties, colors, and shapes vary at different growth stages. (2) The inter-row space of field crops is sometimes filled with weeds or non-crop-related elements, leading to additional problems in image segmentation tasks. (3) The presence of diverse spatial patterns of background (soil, floor coverings) or scenes in greenhouses and open fields due to farmers’ agricultural practices, wetting and drying, and other factors. (4) Diverse and dynamic environmental conditions (summer, winter, dry, wet, fog), geographic location (highlands, lowlands, plains), uncontrolled lighting (sunny, cloudy, daytime, night), and shadows contribute significant challenges to image acquisition and often produce noisy images. This situation we can observe in thermal images; when adjacent pixels between the crop and background are too similar, it is difficult to extract vegetation from the background. (5) The pixel imbalance between the target crop and the background in the image is challenging for automatic segmentation of different objects.

We propose an encoder–decoder architecture based on multi-branch dilated convolutions with traditional convolution and pooling layers as means to address the above constraints. Dilated convolution improves feature extraction performance by maintaining spatial resolution during down sampling tasks [5]. The depth of the network allows the extraction of low-level and high-level features at different depths of the network, thereby improving the accuracy of segmentation with noisy images [6]. The designed network is evaluated on classic pretrained networks such as deeplabv3plusInceptionResNetv2, deeplabv3plusXceptionNet, deeplabv3plusResNet50, and deeplabv3plusMobilenetv2, which have been considered as references. The main contributions of this study are listed as follows. (1) A new CNN architecture is organized into multi-branch parallel blocks for abstracting low-level, medium-level, and high-level features. Furthermore, we concatenate these low-level and high-level features to enrich semantic information, thereby improving segmentation accuracy. (2) The concatenation of dilated convolutional layers with different dilation rates placed in parallel expands the receptivity of multi-scale CNNs by maximizing the spatial feature information, preventing gridding issues, and extracting small regions of the input image more accurately. (3) The network can leverage contextual information and perform semantic segmentation in real time at 60 frames per second (fps) by using minimum GPU memory.

## 2. Related Works

Recently, pixel-level image classification tasks such as scene recognition and semantic segmentation continued to develop and great progress has been made inspired by the combination of deep convolution layers with dilated convolution and pooling layers and substituting fully connected layers [7]. However, training deep learning is associated with various problems, such as gradient instability problems (requiring hyperparameter optimizing), the trend of deep networks that converge with network accuracy beginning to drop significantly, and training error increases [8,9]. A deeper residual-based network, ResNet, identifies low-resolution, high-level features, compensates for the vanishing gradient issue, and demonstrates excellent training results learned from shallow-depth, high-resolution, low-level features [10,11]. MobileNetV2 uses depthwise-separable convolutions and a linear bottleneck with an inverted residual structure, which reduces computational complexity and allows its use in mobile devices [12]. It has been reported that resampling an input image with multiple filters can capture useful image features at multi-level scales with complementary effective fields of view [8]. DeepLabV3plus introduces variable dilation rate-based dilated convolutions to increase receptive field, preserve resolution, and extract multi-scale information. Furthermore, concatenating low-level and high-level feature maps can achieve excellent semantic results [13]. Others utilize data augmentation techniques to preserve the geometry and physical appearance of plants in images and improve leaf segmentation accuracy results [14]. Inception-Resnet-v2 combines convolutional filters of multiple sizes through residual connections, achieving very good performance at relatively low computational cost, without the degradation problems caused by deep structures [15]. XceptionNet achieves even better results on larger datasets, inspired by the inverse sequence of operations in the depthwise-separable convolution (spatial convolution carried out separately over each input channel) preceded by a point-specific convolution and the removal of nonlinearity between convolutional layers [11,16].

## 3. Materials and Methods

This section describes the method proposed in this paper. Section 3.1 describes image acquisition and data strategy. Section 3.2 introduces asymmetric multi-branch module of the ThelR547v1. Section 3.3 presents the dilated (atrous) convolution and overall architecture of ThelR547v1. Section 3.4 details the network’s training parameters. Section 3.5 describes the evaluation metrics for comparative analysis.

### 3.1. Image Acquisition and Dataset Preparation

In this study, Tomato, and lettuce leaves were photographed from several platforms (random—handheld, fixed, aerial—DJI Mini 2, DJI Inspire 2, Da-Jiang Innovations Science and Technology Co., Shenzhen, China), RGB mobile devices (Samsung SCV33, Samsung SCV36, Suwon, Korea; Apple iPhone SE, iPhone 8, Cupertino, CA, USA), RGB cameras, namely Sony (ILCE-6000, DSC-RX100M6, Tokyo, Japan), Nikon Corporation (Nikon D200, Nikon D300, Nikon D700, Nikon D7100, Tokyo, Japan), and LWIR (spectral range 8~14 μm) thermal cameras (Thermo FLEX F50B-ONL, RX450, Nippon Avionics Co., Ltd., Yokohama, Japan). All thermal RGB images were stored with an emissivity of 0.98 98 [17]. All images (thermal RGB, RGB) taken from top and back, close-up full viewing angle, distance (from 0.2 m to 20 m), and simulating the real situation to the greatest extent. The specification and diverse spatial pattern of collected images for dataset preparation are shown in Table 1 and Figure 1, respectively.

A total of 18,664 images (thermal RGB, RGB) resized to 240 × 240 × 3 pixels to confirm the desired network input image size, with a thermal RGB to RGB ratio of 95:5. For our experiments, we collected color and monochrome thermal image datasets (Figure 1), as some thermal cameras use a monochrome filter to visualize noisy images, but sometimes there is still noise on some monochrome thermal images. We then denoised them by thermal image processing software (InfReC Analyzer NS9500STD for F50, Nippon Avionics Co., Ltd.). A time consuming and tedious manual image labeling was performed with a large number of datasets. Image Segmenter (Image Processing and Computer Vision Toolbox, MATLAB R2022a, MathWorks, Natick, MA, USA) was used to classify the pixels of each image into two classes: leaf/leaves/canopy (255) and background (0) and stored in a binary image. A schematic diagram of image dataset preparation and analysis (from dataset preparation to network training, model evaluation, and prediction) is shown in Figure 2a. As shown in Figure 2b, the frequency levels (ratio of leaf to background) of leaf and background pixels in the total image dataset are 70% and 30%, respectively. In this experiment, we use 60% of randomly selected images (images and binary images) for training, 20% for validation, and 20% for testing purposes.

We randomly augment the training dataset to expand the collected dataset and increase the type of variations to avoid model overfitting, reduce sensitivity, improve robustness, and enhance generality and performance. Table 2 and Table 3 show the property of the augmented images and number of image datasets used for deep learning analysis. We augment the training image dataset for performance studies (Aug1) and comparative analysis (Aug2), as shown in Table 3.

We further investigated ThelR547v1 segmentation performance using an independent public dataset of 512 potted *Arabidopsis thaliana* images with 2 classes (leaf and background) downloaded from the IPPN plant phenotype dataset (leaf segmentation challenge component of the CVPPP workshop: CVPPP2017LSC-2017 and CVPPP2017LCC-2017 [2].

### 3.2. Asymmetric Multi-Branch Module

The proposed network is based on asymmetric multi-branch modular type (Figure 3a–d) for achieving high accuracy with less inference time and low latency in real-time scenarios and is shown in Figure 1. We use standard convolution, dilated convolution (standard convolution in the encoding layer), transpose convolution (standard convolution in the decoding layer), and standard convolution in the classification layer. The output of these layers normalizes using Batch (BN), Group (GN), and Instance (IN) layers for channel shuffling. Furthermore, these activate using ReLU and Swish activation layers throughout the network for achieving high accuracy with less inference time and low latency in real-time scenarios. The advantage of these types of structures is that we can add or remove these modules depend on data size, computing capability, and semantic goal complexity.

### 3.3. Image Acquisition and Dataset Preparation

Standard convolution with dilation rate *dr* = 1 is described in Equation (1) [18], as shown in Figure 4a. Dilated convolution is a technique that involves pixel skipping while expanding the kernel (input) by inserting gaps between its consecutive pixels (Chen et al., 2017). Figure 4b,c illustrate the dilated convolution with *dr* = 2 and *dr* = 3, as explained in Equation (2) (modified from Yu and Koltun [18]). It allows us to view the larger receptive field and maintain high resolution of the output image with less computational cost.
(1)$$Cx_{p}=\underset{s+t=p}{\sum }(\left(f\left(s\right)\times k\left(t\right)\right)+b)$$
(2)$$Dx_{p}=\underset{s+t_{r}=p}{\sum }(\left(f\left(s\right)\times k\left(t_{r}\right)\right)+b)$$

Here *f* is a discrete function, *t* is the discrete filter of size (2*r +* 1*)*^2^ of the kernel *k*, *p* represents the position, *f(s)* is the input, *r* is the dilation rate *(r =* 1, 2, 3*,…)*, *t_r_* denotes the kernel size (2*rxd_r_* + 1)^2^, *Cx_p_* denotes output from convolution operation, *Dx_p_* is the output from dilated convolutional operation, and *b* is the bias.

The concatenation layer based on depth, concatenates different level of feature maps along the channel dimensions [19] to increase the feature maps depth information. The concatenation layer based on addition and multiplication add or multiply multiple feature inputs from multiple layers into an element, respectively. Concatenation of dilated convolution using depth function, multiplication function, and add function is defined in Equations (3)–(5).
(3)$$C_{d,p}=Contact\left[{\left(Dx_{p}\right)}_{1},{\left(Dx_{p}\right)}_{2},{\left(Dx_{p}\right)}_{3},\ldots \ldots \right]$$
(4)$$C_{m,p}=Contact\left[{\left(Dx_{p}\right)}_{1}\times {\left(Dx_{p}\right)}_{2}\times {\left(Dx_{p}\right)}_{3},\ldots \ldots \right]$$
(5)$$C_{a,p}=Contact\left[{\left(Dx_{p}\right)}_{1}+{\left(Dx_{p}\right)}_{2}+{\left(Dx_{p}\right)}_{3},\ldots \ldots \right]$$

Transposed convolution operation is described below (modified from Dumoulin and Visin [20])
(6)$$T_{r}Cx_{p}=\underset{s+t=p}{\sum }\left(\left(\frac{f\left(s\right)+2P_{ad}-k\left(t\right)}{s}+1\right)+b\right)$$

Here *P_ad_* is the padding (same), *s* denotes the stride (*s* = 1, 2, 3, 5), *T_r_Cx_p_* is the size of output.

We employ max pooling to reduce network parameters, ensure feature locations, reduce rotation invariance, and overfitting problems [21,22]. The max pooling layer follows the depth-based and addition-based concatenation layer. This layer extracts the maximum value from the outputs of the depth (Equation (3)) and addition-based (Equation (5)) concatenation layers using pooling window sizes of 3 × 3 and 5 × 5, respectively. It can be defined as:(7)$$Max_{p}=maxW\left(t\right)\times C_{d,p}$$
(8)$$Max_{p}=maxW\left(t\right)\times C_{a,p}$$
where *Max* denotes the max pooling layer, and *W(t)* is the pooling size.

We place a normalization layer (Batch, Group, Instance) between the standard convolution/group convolution/dilated convolution/transposed convolution/addition-based concatenation layer and the ReLU/Swish layer, aiming to achieve the following advantages: (1) channel shuffling that stabilizes and increases the learning process by reducing the intra-network covariate shift (ICS); (2) reducing the number of epochs to converge in fewer epochs; (3) learning more non-linear information from the feature map, resulting in qualitative prediction result; and (4) allowing smaller batch sizes to run them on computers with lower computing power and memory.

The most used activation function in this network is ReLU. However, considering the weakness of the ReLU due to the neuron death problem (when ReLU converts the value of each input element less than zero to zero, weights cannot be updated by backpropagation), we placed the Swish layer [23] in the shallow, middle and deep depths of the network. These non-linear activation functions (ReLU, Swish) are applied to the output of the normalization layer (Batch, Group, Instance) and addition-based concatenation layer. This has the following advantages: (1) ReLU prevents the vanishing gradient problem in deep depth of the network layers, allowing the model to learn faster with higher accuracy [24]. (2) It prevents neuron dead issues and introduces strong regularization effects to speed up network convergence, effective for optimization, and generalization.

A 2D crop layer [25] cropping feature maps at a rectangle scale size of [1 1]. The output of this layer then goes through a dropout layer, which randomly turns off 20% of dead neurons to prevent the network from overfitting. We randomly turn off 20% of dead neurons in the middle depth of the encoding and decoding layers to optimize the model learning efficiency. The last convolution layer has two sparse feature maps, each with a pixel size of 240 × 240. The output of the last convolution layer is fed to the Softmax layer through a ReLU activation layer. The output of the Softmax function is a probability score between 0 and 1 for each pixel mapped to the crop and background, respectively. The final layer of the classification layer is binary cross-entropy, which produces a categorical label (leaf/canopy or background) for each pixel and computes the prediction error rate as a loss function. The characteristics of the ThelR547v1 architecture are shown in Table 4.

The network has dived into four layers: encoding layer (shallow depth: step 1 to step 7; middle depth: step 1 to step 8), decoding layer (middle depth: step 9 to step 11; deep depth: step 1 to 8), classification layer (deep depth: step 9 to 12), and output layer (deep depth: step 13 to 14), as shown in Figure 5. In shallow depth (marked in light grey color), middle depth (marked in light red color), and deep depth (marked in light green color) network extracts low-level, medium-level, and high-level features. Summary of the ThelR547v1 configuration is presented in Table 5.

### 3.4. Network Training Parameters

The network was trained on a computer environment MATLAB(R) R2022a with Image Processing Toolbox, Computer Vision Toolbox, Statistics and Machine Learning Toolbox, Deep Learning Toolbox, and Parallel Computing Toolbox. Table 6 presents the hardware and software configuration for deep learning tasks.

The network weight used optimized by an adaptive moment estimation (ADAM) optimizer to solve sparse gradients on noisy datasets [26]. The training parameters applied to train ThelR547v1 were validation frequency: 50; validation patience: inf; mini-batch size: 30/50; batch normalization statistics: population; maximum epoch: 5/10/15/20/25/30; mini-batch size: 12/22/32/42/52; learn rate schedule: piecewise; shuffle: every epoch; initial learn rate: 1 × 10^−3^; learn rate drop period: 10; learn rate drop factor: 0.1; L2 regularization: 1 × 10^−4^; epsilon: 1× 10^−8^; squared gradient decay factor: 0.999; gradient decay factor: 0.9; gradient threshold method: l2norm; gradient threshold: inf; sequence length: longest; sequence padding value: 0; sequence padding direction: right; dispatch in background:0; reset input normalization:1; and output network: last iteration. Due to insufficient computing power, we limit the maximum size of mini-batch to 52.

### 3.5. Evaluation Metrics for Comparative Analysis

There are several performance metrics such as training and validation accuracy (TA and VA show the percentage of correctly classified pixels, %), global accuracy (GA—measuring ratio of correctly classified pixels to the total number of pixels, 1—best value), mean accuracy (MA—measuring the percentage of correctly identified pixels for each class, 1—best value), confusion metrics, loss (TL for training loss and VL for validation loss, 0—best value), pixel/object-wise mean intersection over union index (mean IoU measures the amount of overlap per predicted class, 1—best value), mean weighted IoU (measure the average IoU of each class, 1—best value), BF score (Boundary F1 measures the quality of the predicted boundary with the ground truth boundary, 1—best value), etc. which are used for quantifying ThelR547v1 prediction efficiency. The same performance metrics were evaluated on modified pretrained Deeplabv3plus networks (InceptionResNetv2, XceptionNet, ResNet50, and MobileNetv2) for transfer learning efficiency analysis. However, we visualize network activations to demonstrate how the network transforms the input information to detect features such as colors, edges in shallow layers, and complex features in deep layers. To do this, we compute the network layer activations and visualize the activations at different depths of the network layer. In addition, we use the gradient-weighted class activation mapping (Grad-CAM) proposed by Selvaraju et al. [27] that sums the weights of each feature map from the last convolution layer, assigns each neuron a probability score by computing the gradient (Equation (9)), and produces a heatmap (Equation (10)) with the same size as the feature map to visualize the location of the largest gradient as the output for a given input class.
(9)$$w_{k}^{c}=\overset{Global\;average\;pooling}{\overbrace{\frac{1}{Z}\underset{i}{\sum }\underset{j}{\sum }\underset{\begin{array}{c} Gradients\;via\;\\ backpropagation \end{array}}{\underbrace{\frac{\partial Y^{c}}{\partial A_{i,j}^{k}}}}}}$$
where *c* is the target class. $A_{i,j}^{k}$ represents the activation map. *i*,*j* denote spatial location. *w* is the class weight. *Y^c^* represents probability of the target class *c*. *Z* is the total number of pixels in the feature map.
(10)$$Heatmap_{Grad-Cam}^{c}=\mathrm{ReLU}\underset{linear\;combination}{\underbrace{\left(\underset{k}{\sum }w_{k}^{c}\times A_{i,j}^{k}\right)}}$$

Here ReLU captures features with positive contributions to the target class and output is a heatmap.

## 4. Results

### 4.1. Quantitative Evaluation

The effect of mini-batch size on ThelR547v1 accuracy and loss is investigated, as shown in Figure 6. The maximum value for training and validation accuracy was obtained when the mini-batch size was 52. The minimum value for training and validation loss was also obtained with mini-batch size 52. Hence, the 52nd mini-batch size is chosen as the optimal training parameter. From Figure 6a,b, it is clear that as the mini-batch size increases from 12 to 32, the network performance increases at a higher rate. However, as the mini-batch size increases from 32 to 52, the network performance increases at a lower rate or barely improves.

Next, we investigate the effect of the number of epochs on pixel-level classification of thermal RGB and RGB images by ThelR547v1, Deeplabv3plusInceptionResNetv2, Deeplabv3plusXceptionNet, Deeplabv3plusResNet50, and Deeplabv3plusMobileNetv2 networks. ThelR547v1 achieves a maximum training accuracy of 96.27% and 94.43% using the aug1 and aug2 datasets, respectively, with a mini-batch size of 52 and a maximum epoch of 30, as shown in Figure 7a (aug1) and 7b (aug2). Under the same conditions, Deeplabv3plusInceptionResNetv2 achieves the nearest training accuracy of 96.15% and 94.00% using aug1 and aug2 datasets, respectively. ThelR547v1 also achieves maximum validation accuracy of 95.88% and 95.31%, respectively, as shown in Figure 7c (aug1) and 7d (aug2).

ThelR547v1 shows higher mini-batch accuracy of 96.27% and 95.52% using aug1 (Figure 7e) and aug2 (Figure 7f) datasets, respectively. Deeplabv3plusInceptionResNetv2 demonstrated the closest mini-batch accuracy of 96.15% and 95.20% using aug1 (Figure 7e) and aug2 (Figure 7f) datasets, respectively. From Figure 7g (aug1), we can see that both ThelR547v1 and Deeplabv3plusInceptionResNetv2 achieve a minimum training loss of 0.09. However, for the aug2 dataset (Figure 7h), ThelR547v1 achieves a minimal training loss of 0.13, but Deeplabv3plusInceptionResNetv2 achieves a slightly higher training loss of 0.14 under the same conditions. The minimum validation losses of 0.10 (Figure 7i) and 0.11 (Figure 7j) are observed on the aug1 and aug2 datasets of ThelR547v1 and Deeplabv3plusInceptionResNetv2, respectively. In addition, ThelR547v1 and Deeplabv3plusInceptionResNetv2 achieve minimum mini-batch losses of 0.90 and 0.10 using aug1 (Figure 7k) and aug2 (Figure 7l) datasets, respectively.

However, Figure 7a,c,e,g,i,k and Figure 7b,d,f,h,j,l indicate that network performance degrades with the aug2 dataset. This response is due to the addition of noise by increasing dataset through augmentation, as described in Table 3. Finally, it is concluded that the aug1 dataset with a mini-batch size of 52 and a maximum epoch of 30 is effective for network training.

The order of error rate for each class (leaf and background) is shown in Figure 8 and Appendix A. It can be seen from the confusion chart in Figure 8a that ThelR547v1 achieves higher classification accuracies of 97.99%, 97.98%, and 98.01% for leaf with training (Figure 8a(i)), validation (Figure 8a(ii)), and test (Figure 8a(iii)) datasets. However, Deeplabv3plusInceptionResNetv2 has a slightly higher classification accuracy of 98.08% for leaf with training (Figure 8b(i)), validation (Figure 8b(ii)), and test (Figure 8b(iii)) datasets. Furthermore, ThelR547v1 exhibits the highest classification accuracy during training (Figure 8a(i)), validation (Figure 8a(ii)), and test (Figure 8a(iii)) background class, with 91.37%, 91.17%, and 91.20%, respectively.

Figure 9 shows that ThelR547v1 achieved the highest GA (0.96), MA (0.95), meanIoU (0.91), weightedIoU (0.92), and meanBFScore (0.89) using a test dataset (12,442 images). Deeplabv3plusInceptionresnetv2 demonstrates similar meanBFScore (0.89). Deeplabv3plusXceptionnet and Deeplabv3plusResNet50 score the lowest meanIoU (0.89). A higher value indicates better network performance.

Table 7 shows the test results of GA, IoU, and meanBFScore against each individual class (leaf and background).

ThelR547v1 has a minimum training parameters of 5.4 M and a moderate network size of 15.4 Mb (Table 8), thus making ThelR547v1 easy to deploy in real-time applications for autonomous semantic segmentation tasks.

### 4.2. Qualitative Evaluation

In Figure 10 and Figure 11, we show some examples of qualitative results based on semantic challenges. The first row in Figure 10a and Figure 11a shows independently collected thermal color and monochrome RGB (tomato) and RGB test image (tomato, lettuce) datasets under various conditions (day/night time), respectively. The second to fourth row in Figure 10b–f and Figure 11b–f display the networks’ (ThelR547v1, Deeplabv3plusInceptionresnetv2, Deeplabv3plusXceptionNet, Deeplabv3plusResNet50, and Deeplabv3plusMobilenetv2) predicted ground truth data (binary). These results show that ThelR547v1 is capable of correctly estimating class boundaries and excluding different types of background noise (soil, plant factory ground, windows, cover sheet, humans, narrow-leaf grass, etc.) better than Deeplabv3plusInceptionresnetv2, Deeplabv3plusXceptionNet, Deeplabv3plusResNet50, and Deeplabv3plusMobilenetv2. This experiment shows that ThelR547v1 can predict any combination of images with higher accuracy. Therefore, this enables the network to be used with any type of thermal imaging camera in a real-world environment.

#### 4.2.1. Transfer Learning Performance Evaluation

In this experiment, we show the transfer learning performance of ThelR547v1, Deeplabv3plusInceptionresnetv2, Deeplabv3plusXceptionNet, Deeplabv3plusResNet50, and Deeplabv3plusMobilenetv2 trained with the aug1 dataset. Figure 12a(i),b(i),c(i),d(i) display a low-resolution RGB image from broad-sized to narrow-sized leaves. Manual ground truth data (binary) in Figure 12a(ii),b(ii),c(ii),d(ii) show two semantic classes (leaf and background). The differences in ground truth data (binary) predicted by the network before and after training with the IPPN plant phenotype RGB image dataset are shown in Figure 12a(iii,v,vii,ix,xi),b(iii,v,vii,ix,xi),c(iii,v,vii,ix,xi),d(iii,v,vii,ix,xi) and Figure 12a(vi,viii,x,xii),b(vi,viii,x,xii),c(vi,viii,x,xii),d(vi,viii,x,xii), respectively. From Figure 12a(iii,v,vii,ix,xi),b(iii,v,vii,ix,xi),c(iii,v,vii,ix,xi),d(iii,v,vii,ix,xi), it is clear that before training Deeplabv3plusInceptionresnetv2, Deeplabv3plusXceptionNet, Deeplabv3plusResNet50, and Deeplabv3plusMobilenetv2 networks misclassified leaf and background more than ThelR547v1 network. However, ThelR547v1, after training, classifies and splits both broad-sized and narrow-sized leaves from their background with higher accuracy, as shown in Figure 12a(iv),b(iv),c(iv),d(iv) compared with Deeplabv3plusInceptionresnetv2, Deeplabv3plusXceptionNet, Deeplabv3plusResNet50, and Deeplabv3plusMobilenetv2 in Figure 12a(vi,viii,x,xii),b(vi,viii,x,xii),c(vi,viii,x,xii),d(vi,viii,x,xii), respectively.

#### 4.2.2. Activated Feature Map Visualization

Examples of activation feature maps of ThelR547v1 at different depths (shallow, middle, and deep) for a test image (lettuce, RGB) are shown in Figure 13. The weight (3 × 3 × 3 × 64) and bias (1 × 1 × 64) as learnable parameters of the first convolution layer (standard) in shallow depth has a feature map size of 240 × 240 with 64 filters (Figure 13a). After that, the feature map size is divided into two groups: 30 × 15 and 64 filters, as shown in Figure 13b (weights and biases are 3 × 3 × 32 × 64 and 1 × 1 × 64, respectively) and 15 × 30, with 64 filters (weights and biases are 2 × 3 × 32 × 64 and 1 × 1 × 64, respectively), as shown in Figure 13c and feed them to standard convolution layers. As shown in Figure 13d, in the encoding layer at middle depth, the feature map size of the first dilated convolution layer is 240 × 240, where 16 filters with learnable parameter weights and biases are 3× 3 × 64 × 16 and 1 × 1 × 16, respectively. The feature map size (15 × 15 × 64), weights (3 × 2 × 64 × 64), and biases (1 × 1 × 64) of the middle depth dilated convolution layer are shown in Figure 13e. The last dilated convolution layer in the middle depth (Figure 13f) has a feature map of size 2 × 2 with 32 filters and weights and biases of 3 × 2 × 32 × 32 and 1 × 1 × 32, respectively.

In the decoding layer of the middle depth, as shown in Figure 13g, the feature map size of the first transposed convolution layer is 2 × 2 with 64 filters, and the weights and biases of the learnable parameters are 2 × 2 × 64 × 192 and 1 × 1 × 64, respectively. In deep depth, as shown in Figure 13h, the feature map produced by the transposed convolution layer is 8 × 8 in size with 64 filters, and the weights and biases of the learnable parameters are 3 × 3 × 64 × 128 and 1 × 1 × 128, respectively. The last transposed convolution at the deep depth of the decoding layer produced a feature map size of 240 × 240 with 32 filters (Figure 13i) with the weight and bias of learnable parameters being 3 × 3 × 32 × 64 and 1 × 1 × 32, respectively.

Furthermore, in the deeper depth of the classification layer (Figure 13j), the feature map produced by the standard convolution is 240 x 240 with 64 filters with the weight and bias of 3 × 3 × 152 × 64 and 1 × 1 × 64, respectively. The output feature map (240 × 240 × 2) of the final standard convolution of the deep depth classification layer is shown in Figure 13k. The weights and biases of this layer are 3 × 3 × 16 × 2 and 1 × 1 × 2, respectively. The softmax in the classification layer shows probability outputs ranging from 0 (black) to 1 (white), as shown in Figure 13l. The pixel classification layer uses a cross-entropy loss function to predict the output, which shows the highest and lowest learnable features in white (leaf) and black (background), as shown in Figure 13m.

#### 4.2.3. Gradient-Based Class Activation Heatmap (Grad-CAM) Visualization in the Different Layers

The gradient-weighted class activation mapping (Grad-CAM) technique captures more abstract and low-level features, such as edges, as shown in Figure 14 (thermal RGB) and Figure 15 (RGB). At shallow depths (Figure 14b), ThelR547v1 mainly focuses on background regions (yellow) associated with leaf class and lightly focuses on leaf (light sky-blue color) associated with background class. In middle depth (Figure 14c), the network focuses on image regions that are related to the leaf and background classes but does not necessarily belong to it, but the network in the deep depth, as shown in Figure 14d, mostly focuses on the ground truth of the leaf and background classes. Finally, in the classification layer shown in Figure 14e, the network produces a deep depth coarse localization map that focuses on high-level features such as patterns, thereby representing the most important ground truth regions of the leaf and background classes in the image.

For the RGB image shown in Figure 15a, the shallow depth of the ThelR547v1 mainly focuses on the image regions related to leaf but not on the background class (Figure 15b). In the middle depth, as shown in Figure 15c, the network focuses on image regions that are related to the leaf and background classes but do not necessarily belong to it. As shown in Figure 15d, network in deep depth mostly focuses on leaf and background classes. Finally, in the classification layer shown in Figure 15e, the network produces a deep depth coarse localization map that focuses on high-level features such as patterns, thereby representing the most important ground truth regions of the leaf and background classes in the image.

### 4.3. Crop Monitoring Performance

For this experiment, we collected 250 independent datasets of thermal RGB and RGB images between October 2021 and October 2022. As shown in Figure 16a, in the case of thermal images, the total pixels predicted by ThelR547v1 for leaf/canopy regions are highly correlated (R^2^ = 0.8607) with data extracted from manual grand truth (binary) images, but in the case of RGB images, this correlation represents 0.9598 (R^2^), as shown in Figure 16b.

The minimum temperature predicted by ThelR547v1 is absolute correlated (R^2^ = 0.9998) with data extracted from the artificial ground truth (binary) images and shown in Figure 17a. Figure 17b exhibits almost similar performance (R^2^ = 0.9487) for estimating leaf/leaf canopy maximum temperature. However, the reason for this achievement is as follows: The plant factory located at Ehime prefecture, Japan had a stable night-time minimum temperature in October. However, due to the dynamic weather conditions in October (sunny, cloudy, rainy), the daytime maximum temperature changes rapidly. This may be behind thermal image noise, which makes predictions difficult, resulting in a slightly poorer correlation of leaf/canopy maximum temperatures predicted by ThelR547v1 with data extracted from manual large real (binary) images.

## 5. Discussion

In our study, we propose a combination of standard convolutions and dilated convolutions with different dilation rates to achieve high-quality local and global segmentation results in real-time scenes. Since manual segmentation of low-resolution images is a challenging task, inconsistencies appear on the boundaries of leaves or thin sections of leaves when they overlap with something other than the target plant. Despite test images with various complexities such as shadows, diffuse light, high contrast, low contrast, thinner leaves, presence of people, machines or other devices/machines, or different interpretations of plant parts, the overall mean accuracy, mean IoU and mean BF scores of TheRnet547v1 are over 0.90, and the overall difference in metrics is small. The remaining limitations and future research directions that we will continue to use are as follows:The training time of ThelR547v1 is 4 times that of Deeplabv3plusInceptionResNetv2, 9.5 times that of Deeplabv3plusXceptionNet, 9.2 times that of Deeplabv3plusResNet50, and 12.2 times that of Deeplabv3plusMobilenetv2. In our future work, we plan to optimize the number of dilations rate of the of the dilated convolution layer and optimize the network hyperparameters.Due to the lack of large datasets, the proposed network is sometimes unable to distinguish the boundaries between the leaves/canopy and background in some cases. To solve this, we plan to continue the collection of more challenging datasets (complex background, environmental noise, various stages of plant growth, cultivation media such as soil and hydroponic conditions such as greenhouses or open fields) and extensive development of manual ground truth (binary) data for training the network.

## 6. Conclusions

We have presented ThelR547v1 by aiming at the problem of separation of target class (leaf) from their background under any environment condition throughout the year. The highly compact network produces high-quality ground truth data that can potentially eliminate manual post-processing steps for separating leaves/canopy from the background. Even with imbalanced data per class, ThelR547v1 can accurately localize object boundaries (leaf and background) with balanced accuracy compared to other state-of-the-art models, such as Deeplabv3plusInceptionResNetv2, Deeplabv3plusXceptionNet, Deeplabv3plusResNet50, and Deeplabv3plusMobilenetv2. This paper also investigated the end-to-end practical semantic segmentation of both thermal RGB and RGB images collected randomly from a plant factory and open field conditions and found that ThelR547v1 performs well with the manually segmented dataset for measuring the pixel area of the leaf/canopy and temperature stress of the crops.

## Figures and Tables

**Figure 1 sensors-22-08807-f001:**
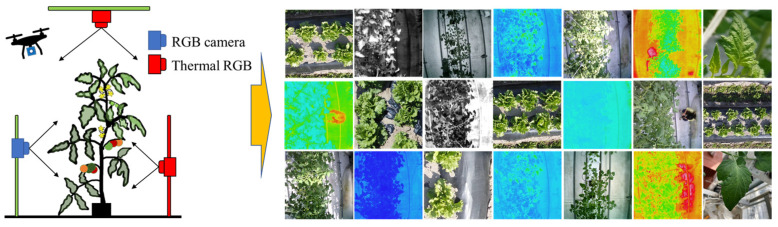
Diverse spatial patterns of input images.

**Figure 2 sensors-22-08807-f002:**
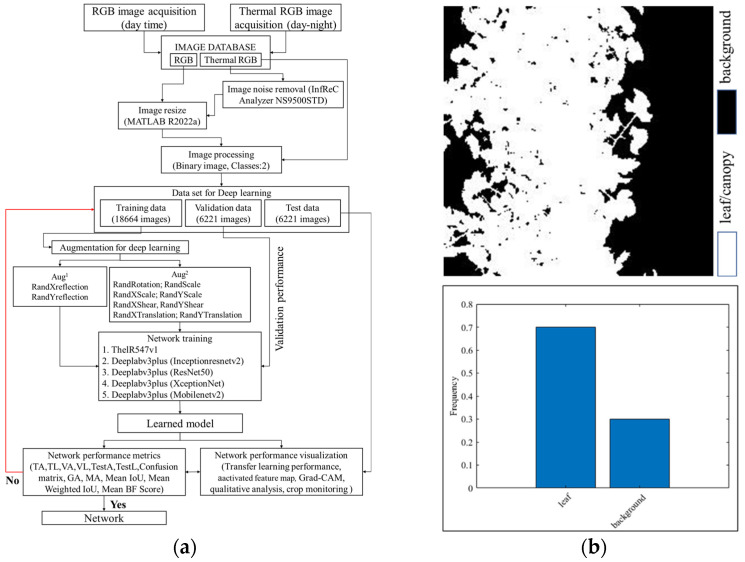
Image dataset preparation approach: (**a**) schematic diagram and (**b**) frequency level of the annotated classes.

**Figure 3 sensors-22-08807-f003:**
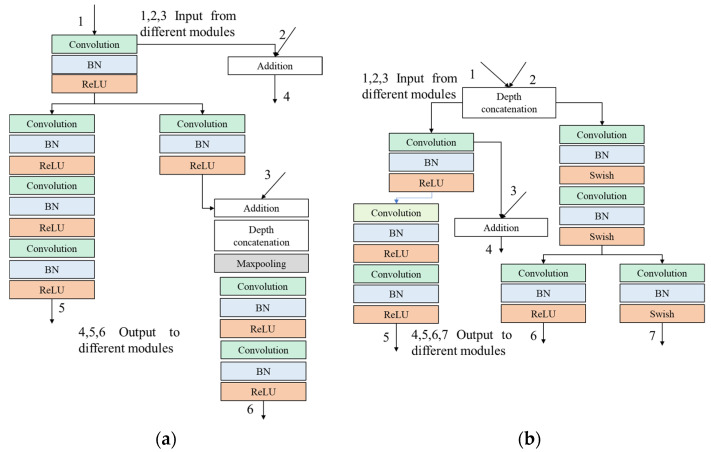
Examples of the multi-branch architecture in different depths of the network: (**a**) shallow depth (encoding layer); (**b**) middle depth (decoding layer); (**c**) deep depth (decoding layer); (**d**) deep depth (classification layer).

**Figure 4 sensors-22-08807-f004:**
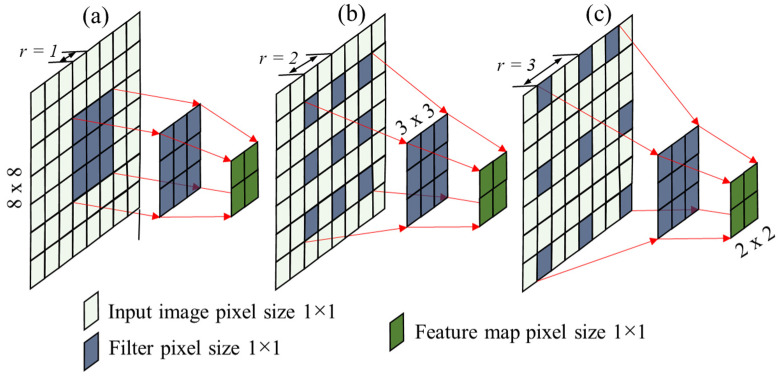
Examples of convolution: (**a**) standard convolution and (**b**,**c**) dilated convolution.

**Figure 5 sensors-22-08807-f005:**
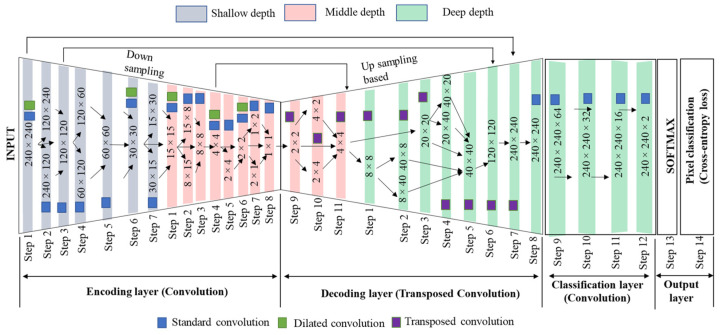
Layer spatial size-based approach of the proposed ThelR547v1 network.

**Figure 6 sensors-22-08807-f006:**
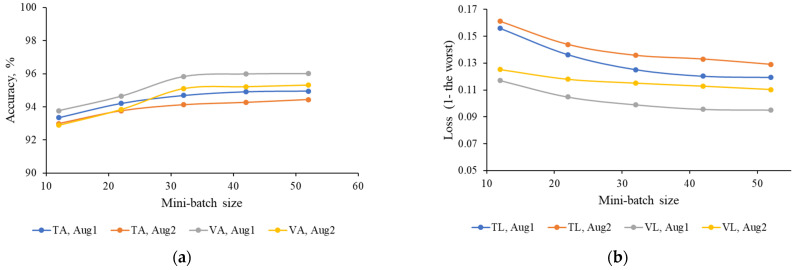
Relationship between mini-batch and network performance. (**a**) Mini-batch size vs. accuracy. (**b**) Mini-batch size vs. loss.

**Figure 7 sensors-22-08807-f007:**
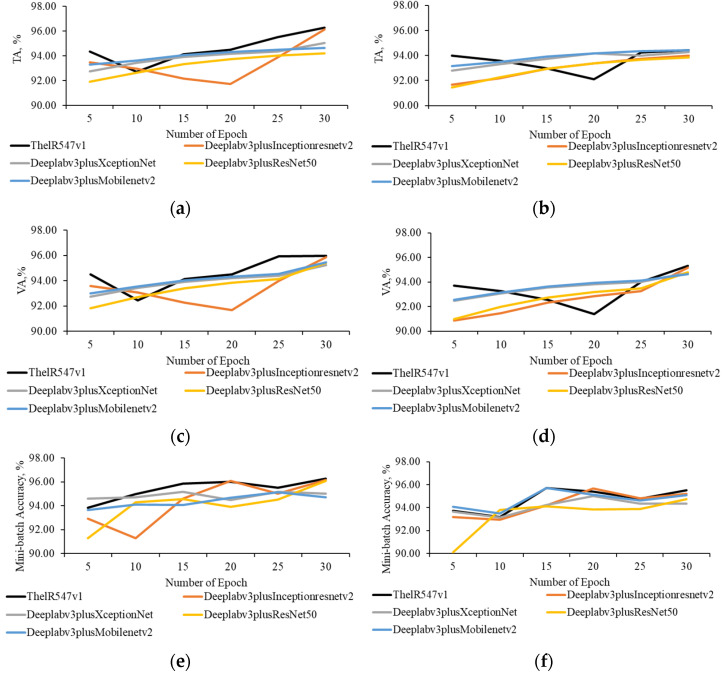
Relationship between number of epoch and network performance using aug1 and aug2 dataset. (**a**) Aug1 dataset. (**b**) Aug2 dataset. (**c**) Aug1 dataset. (**d**) Aug2 dataset. (**e**) Aug1 dataset. (**f**) Aug2 dataset. (**g**) Aug1 dataset. (**h**) Aug2 dataset. (**i**) Aug1 dataset. (**j**) Aug2 dataset. (**k**) Aug1 dataset. (**l**) Aug2 dataset.

**Figure 8 sensors-22-08807-f008:**
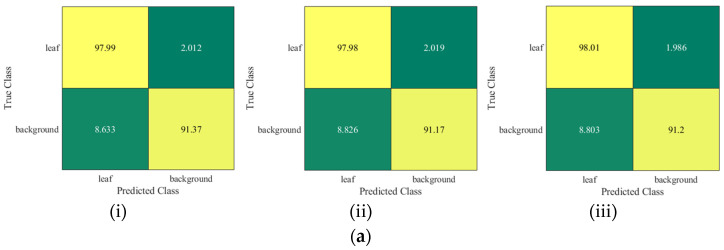
Confusion matrix with training, validation, and testing dataset. (**a**) ThelR547v1: (i) training performance; (ii) validation performance; (iii) test performance. (**b**) Deeplabv3plusInceptionResNetv2: (i) training performance; (ii) validation performance; (iii) test performance.

**Figure 9 sensors-22-08807-f009:**
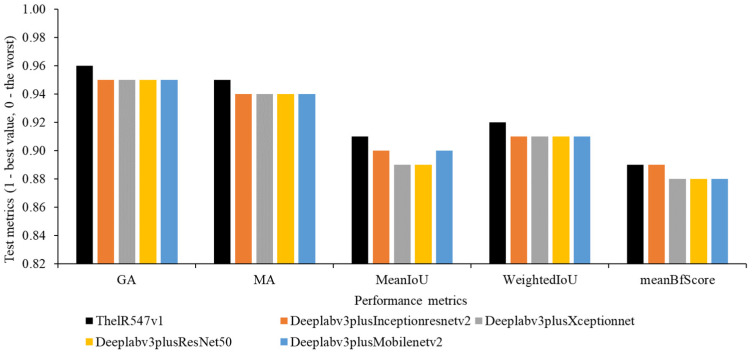
Visualization of the network overall performance.

**Figure 10 sensors-22-08807-f010:**
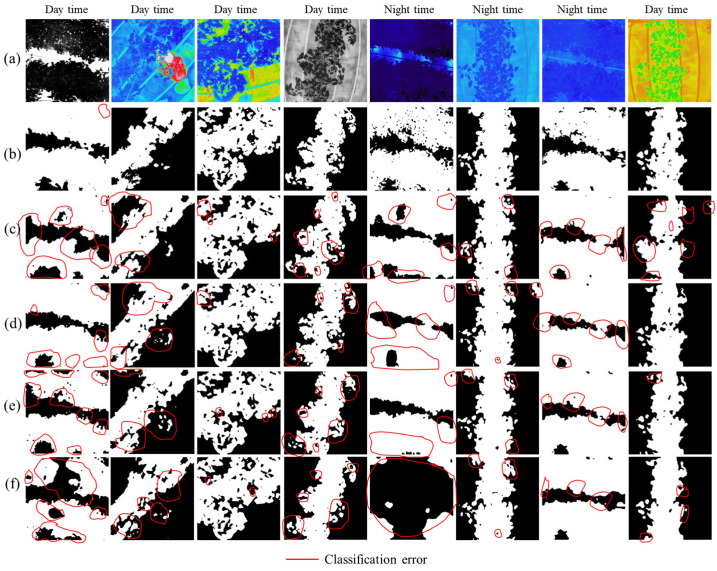
Visualization of the challenges for semantic segmentation of crops (thermal RGB) grown in plant factory: (**a**) thermal color and monochrome RGB image of tomato; (**b**) ThelR547v1 predicted ground truth image; (**c**) Deeplabv3plusInceptionresnetv2 predicted ground truth image; (**d**) Deeplabv3plusXceptionNet predicted ground truth image; (**e**) Deeplabv3plusResNet50 predicted ground truth image; (**f**) Deeplabv3plusMobilenetv2 predicted ground truth image.

**Figure 11 sensors-22-08807-f011:**
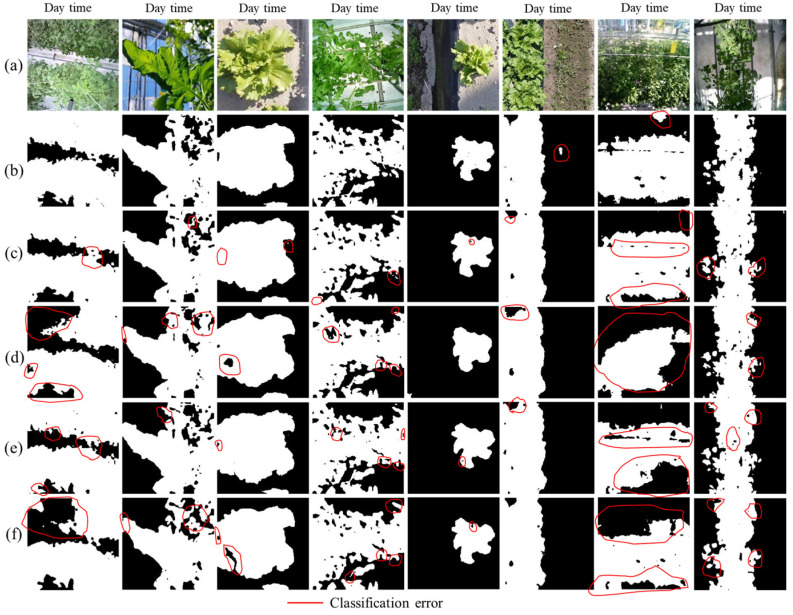
Visualization of the challenges for semantic segmentation of crops (RGB) grown in plant factory and open field: (**a**) RGB image of tomato and lettuce; (**b**) ThelR547v1 predicted ground truth image; (**c**) Deeplabv3plusInceptionresnetv2 predicted ground truth image; (**d**) Deeplabv3plusXceptionNet predicted ground truth image; (**e**) Deeplabv3plusResNet50 predicted ground truth image; (**f**) Deeplabv3plusMobilenetv2 predicted ground truth image.

**Figure 12 sensors-22-08807-f012:**
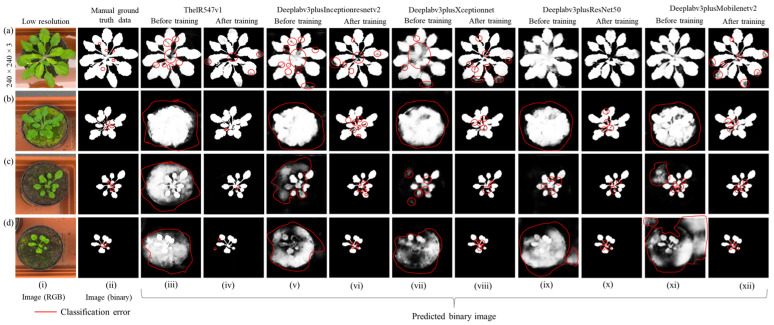
Visualized semantic segmentation error using CVPPP2017LCC/LSC-2017 data: (**a**) broad-sized leaves of *Arabidopsis thaliana*; (**b**) medium-sized leaves of *Arabidopsis thaliana*; (**c**) narrow-leaves of *Arabidopsis thaliana*; (**d**) narrowest-leaves of *Arabidopsis thaliana*.

**Figure 13 sensors-22-08807-f013:**
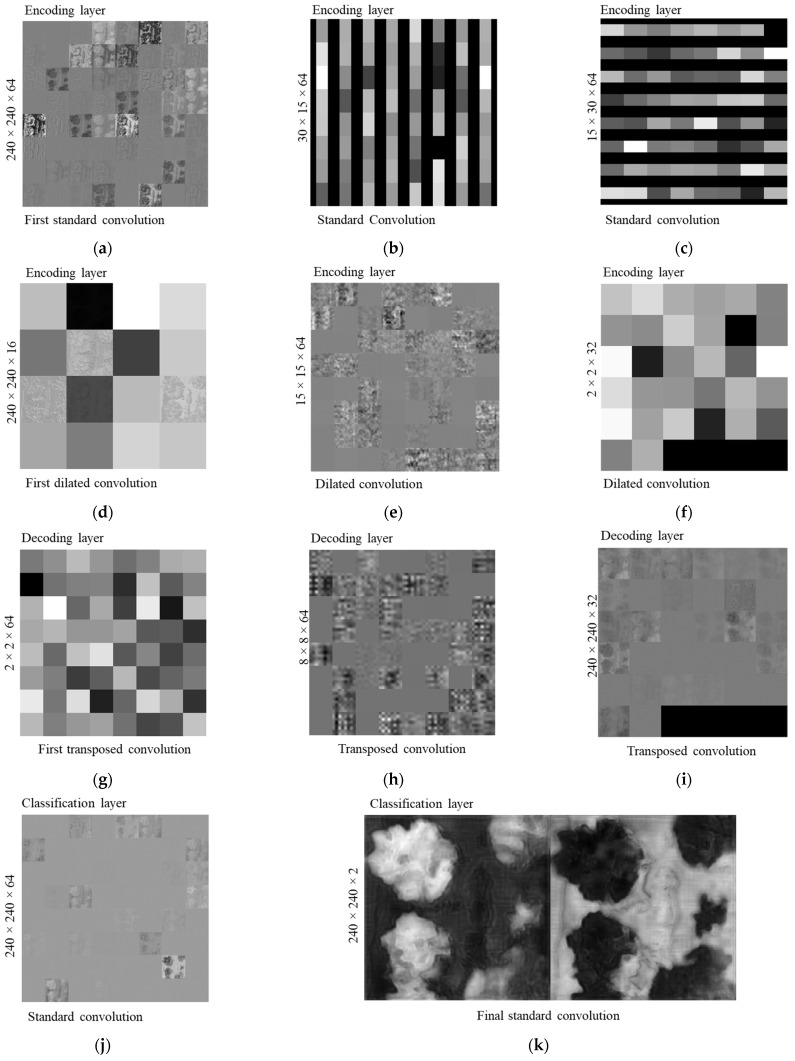
Visualization of the network activated feature maps. (**a**) Shallow depth. (**b**) Middle depth. (**c**) Deep depth. (**d**) Shallow depth. (**e**) Middle depth. (**f**) Middle depth. (**g**) Middle depth. (**h**) Deep depth. (**i**) Deep depth. (**j**) Deep depth. (**k**) Deep depth. (**l**) Deep depth. (**m**) Deep depth.

**Figure 14 sensors-22-08807-f014:**
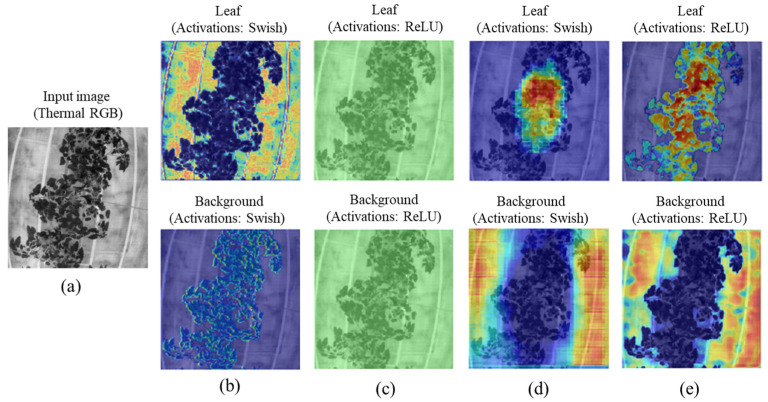
Visualization of the Grad-CAM in different depth of the network: (**a**) monochrome thermal RGB image; (**b**) Grad-CAM visualization at shallow depth of the encoding layer; (**c**) Grad-CAM visualization at middle depth of the encoding layer; (**d**) Grad-CAM visualization at deep depth of the decoding layer; (**e**) Grad-CAM visualization at deep depth of the classification layer.

**Figure 15 sensors-22-08807-f015:**
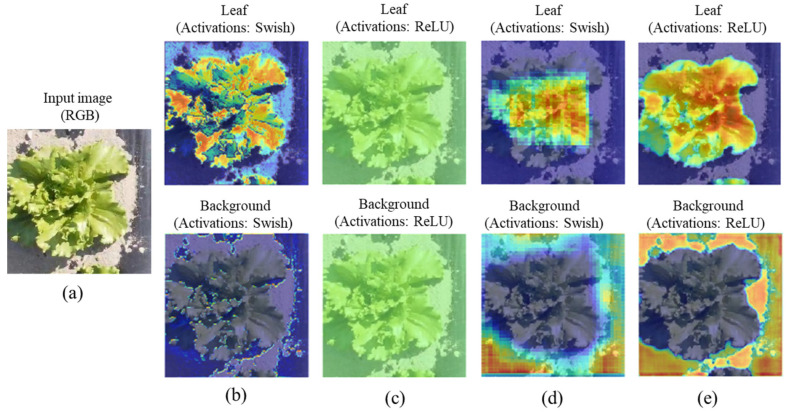
Visualization of the Grad-CAM in different depth of the network: (**a**) RGB image; (**b**) Grad-CAM visualization at shallow depth of the encoding layer; (**c**) Grad-CAM visualization at middle depth of the encoding layer; (**d**) Grad-CAM visualization at deep depth of the decoding layer; (**e**) Grad-CAM visualization at deep depth of the classification layer.

**Figure 16 sensors-22-08807-f016:**
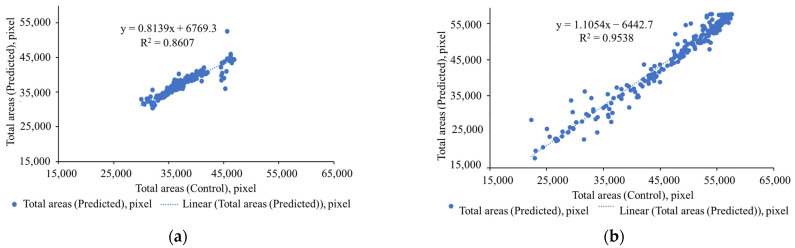
Comparative analysis of leaf/canopy pixel monitoring: (**a**) thermal RGB and (**b**) RGB.

**Figure 17 sensors-22-08807-f017:**
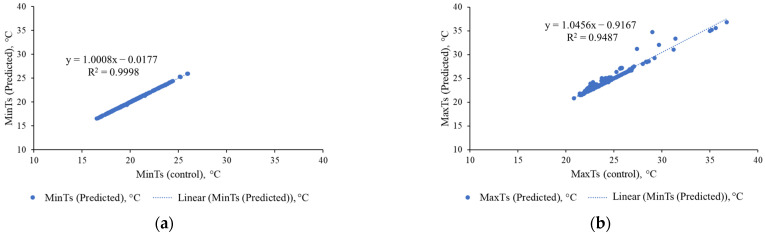
Comparative analysis of leaf/canopy temperature monitoring: (**a**) minimum temperature, thermal RGB; (**b**) maximum temperature, thermal RGB.

**Table 1 sensors-22-08807-t001:** Specification of collected images.

Type of Imaging Platform	Aspect Ratio	Pixel Resolution	Imaging Condition
RGB	1:1, 16:9, 4:3	6000 × 4000, 4928 × 3264, 4288 × 2848, 4256 × 2832, 4032 × 1960, 3872 × 2592, 3648 × 3648, 3024 × 4032, 3024 × 3024, 2160 × 2160, 1960 × 4032, 720 × 720	Heterogeneous illumination and background variation
Thermal RGB	1:1	240 × 240, 480 × 360	

**Table 2 sensors-22-08807-t002:** Augmentation option for comparative analysis.

Augmentation Option	
Aug1	Aug2
RandXreflection: 1	RandXreflection: 1
RandYreflection: 1	RandYreflection: 1
	RandRotation: [−10 10]
	RandScale: [1 1]
	RandXScale: [1 1.2]
	RandYScale: [1 1.2]
	RandXShare: [−2 2]
	RandYShare: [−2 2]
	RandXTranslation: [−3 3]
	RandYTranslation: [−3 3]

**Table 3 sensors-22-08807-t003:** The number of image datasets for deep learning analysis.

Original Dataset	Image Dataset	Binary Dataset	Total Number of Images	Augmented Dataset	
				Aug1	Aug2
Training	18,664	18,664	37,328	111,984	410,608
validation	6221	6221	12,442	-	-
Test	6221	6221	12,442	-	-

**Table 4 sensors-22-08807-t004:** Characteristics of the ThelR547v1 network architecture.

Layers Name	Number of Layers	Layers Name	Number of Layers
Image input	1	ReLU	143
Convolution	92	Swish	23
Group convolution	2	Concatenation (depth)	16
Dilated convolution	15	Concatenation (addition)	19
Transposed convolution	59	Concatenation (multiplication)	2
Batch normalization	157	Dropout	3
Group normalization	4	Crop 2D	1
Instance normalization	4	Softmax	1
Max pooling	4	Binary-cross entropy	1

**Table 5 sensors-22-08807-t005:** Summary of the ThelR547v1 configuration.

Network Depth	Network Layers	Filter Size	Dilate Rate	Stride Rate	Max Pooling Size
Shallow depth	Encoding	3 × 3; 1 × 3; 3 × 1; 2 × 3	1,1; 2,2	1,1;1,2; 2,1; 2,2	3 × 3
Middle depth	Decoding	3 × 3; 1 × 3; 2 × 2; 3 × 2; 2 × 3	1,1; 2,2; 3,3	1,1; 2,1; 1,2; 2,2	5 × 5
Deep depth	Decoding	3 × 3; 2 × 2; 3 × 1; 2 × 3	1,1	1,1; 2,2; 5,5; 1,5; 5,1; 1,2; 2,1; 3,3	-
	Classification	3 × 3	1,1	1,1	-

**Table 6 sensors-22-08807-t006:** Configuration parameters.

Configuration Item	Parameter Value
CPU	Intel(R) CPU Core (TM) i9-10850K CPU @ 3.60 GHz (Cores 10) × 20 processors
GPU	NVIDIA GeForce RTX 3060 PCI-E with GDDR6 (1.78 GHz), 2 × 12 GB
Compute Unified Device Architecture (CUDA)	3584 cores, 12.74 TFLOPS
Software acceleration	Nvidia CUDA 11.2
Memory	128 GB
Hard disk	Samsung SSD 970 EVO Plus (500 GB)
Operating system	Windows 10 pro (64-bit)

**Table 7 sensors-22-08807-t007:** Network overall performance for each individual class prediction (leaf and background).

Network Name	Leaf			Background		
	GA	IoU	Mean BfScore	GA	IoU	Mean BfScore
ThelR547v1	0.98	0.94	0.93	0.91	0.87	0.86
Deeplabv3plusInceptionresnetv2	0.98	0.94	0.93	0.90	0.86	0.85
Deeplabv3plusXceptionNet	0.97	0.93	0.92	0.89	0.84	0.83
Deeplabv3plusResNet50	0.97	0.93	0.92	0.89	0.85	0.83
Deeplabv3plusMobilenetv2	0.95	0.93	0.92	0.90	0.85	0.84

**Table 8 sensors-22-08807-t008:** Comparison between the proposed ThelR547v1 and other pretrained networks.

Network Name	Network Size, Mb	Number of Layers	Number of Parameters, M	Total Number of Connections
ThelR547v1	15.5	547	5.4	1222
Deeplabv3plusInceptionResNetv2	204	853	71.1	1912
Deeplabv3plusXceptionNet	60.2	205	27.6	444
Deeplabv3plusResNet50	104	206	43.9	454
Deeplabv3plusMobileNetv2	7.4	186	6.7	402

## Data Availability

Data will be available upon completion of investigation and publication of the report upon request to the corresponding author. Any request will be reviewed and approved by the sponsor, Ehime University, intellectual property department, and staff based on the absence of competing interest. Once approved, data can be transferred after signing of a data access agreement and confidentiality agreement.

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
