# Peer review of "ThelR547v1—An Asymmetric Dilated Convolutional Neural Network for Real-time Semantic Segmentation of Horticultural Crops"

_sensors, 2022, doi:10.3390/s22228807_

Round 1

Reviewer 1 Report

This paper presents a deep learning architecture for segmenting crop pixel and other pixels. This study is essentially the need of the hour and the architecture uses multi-branch parallel blocks for abstracting low-level, medium-level and high-level features.

The authors have presented a good contribution to the research, however, it is worth it to justify the variations in the architecture of the existing studies with the proposed method.

In Figure 7. Relationship between number of epoch and network performance using aug1 and aug2 328 dataset (b) and (d) shows a decrease in accuracy at epoch 20. What may be the reason?

In Figure 8. Confusion matrix with training, validation, and testing dataset can be given as the single table with accuracy, precision F1 Score etc

The ablation study details are missing.

Author Response

In Figure 7. Relationship between number of epoch and network performance using aug1 and aug2 328 dataset (b) and (d) shows a decrease in accuracy at epoch 20. What may be the reason?

Answer:  We used Aug 2 data to increase the dataset quantitatively. However, it adds more noise to the input image. For this cause, all networks show the same performance. However, TheRnet547v1 shows decrease in accuracy at epoch 20 and increase accuracy with higher mini-batch size means, that our proposed network will perform well with mini-batch size more than 20 to 52. Due to insufficient computing power, we limit the maximum size of mini-batch to 52. So, in this moment it is difficult to say under which maximum value the performance will degrade again.

In Figure 8. Confusion matrix with training, validation, and testing dataset can be given as the single table with accuracy, precision F1 Score etc

We have revised the figure by considering the suggestions of all reviewers. 8 (from lines 364 to 370). We have added Appendix A to demonstrate the network performance of Deeplabv3plusXceptionnet, Deeplabv3plusResNet50, and Deeplabv3plusMobilenetv2.

The ablation study details are missing.

Answer:  This is a time-consuming process. In this study, we mainly focus on dataset preparation strategies, network architectures, and performance strategies based on real-world environments. We hope that in our next study, we will include ablation studies.

Reviewer 2 Report

In this paper, the deep learning architecture ThelR547v1 (Thermal leaf RGB 547 21 layers version 1) is proposed to segment white light (RGB) and infrared (thermal) images of agricultural products. Its performance has been compared with the state-of-the-art methods in terms of accuracy and loss. Through analysis on data-bases and experimental data, it has been shown that it outweighs the other methods although the consumed time is longer. The paper is well-written and can be considered publishable in MDPI-Sensors provided that the following concerns are addressed:

Major:

1- The term frequency in Fig. 2(b) is not clear nor does it have a unit. Does it indicate the ratio of leaf to background in all images?

2- The mini-batch size 52 has been chosen because of the resulted best accuracy and loss. As shown in Fig. 6, only the interval 30 and 55 has been considered. Please elaborate on the reason of choosing this interval for performance evaluation.

3- Please explain why the colormaps of the thermal images in Figure 10 are different and whether they influence the segmentation performance.

4- I disagree with the statement in lines 392-394 that TheRnet547v1 before training does not misclassify leaf and background. The counter example is Figure 12 (iii)(b-d). It is also hard to say whether TheRnet547v1 outperform the other methods since the evaluation is subjective and other methods look to me as good as the TheRnet547v1.

Minor:

5- Line 18: … such as complex…

6- Line 57: by maintaining…

7- Line 162: these types of structures…

8- Line 249: The correct Table number must be 6. Also, the CPU specifications has been repeated in the last row.

9- Figure 6(b), 7(h, j, l) and 9 in the y-label: 1- the worst

10- There is a lot of unnecessary and repetitive features in Figure 8: Since the value of each segment in each matrix is given, the colorbars sound unnecessary. Also, the title of subfigures Normalized Confusion Matrix (%) can be removed and mentioned once in the figure caption.

11- Line 360: …training parameters of…

12- In caption of Figure 7, it should be mentioned which subfigures are associated with aug1 and which with aug2. Just mentioning it in body text is not sufficient: the figures must be independently readable.

13- Similar to previous comment, in caption of Figure 8 it should be mentioned what (i), (ii) and (iii) refer to.

14- Similar to previous comment, in Figures 10-15 the subfigures (a), (b), (c) etc. should be introduced and described in the captions.

15- Line 396: compared with.

16- Line 470: ground truth

17- Line 517: open field condition

Author Response

Major:

1- The term frequency in Fig. 2(b) is not clear nor does it have a unit. Does it indicate the ratio of leaf to background in all images?

Answer: Modified. Please check Line 143:

2- The mini-batch size 52 has been chosen because of the resulted best accuracy and loss. As shown in Fig. 6, only the interval 30 and 55 has been considered. Please elaborate on the reason of choosing this interval for performance evaluation.

Answer: we added more dataset and modified the Fig. 6 in Line 311. We also modified the paragraph 4.1 by adding text. Please check Line 307 to 310.  Due to insufficient computing power, we limit the maximum size of mini-batch to 52.

3- Please explain why the colormaps of the thermal images in Figure 10 are different and whether they influence the segmentation performance.

Answer: We modified paragraphs 3.1 and4.2 by adding text. Line 132  to 137; Line 389; Line 397 to 400.  We want to use this network with real-world situation with any type of thermal or RGB camera.

4- I disagree with the statement in lines 392-394 that TheRnet547v1 before training does not misclassify leaf and background. The counter example is Figure 12 (iii)(b-d). It is also hard to say whether TheRnet547v1 outperform the other methods since the evaluation is subjective and other methods look to me as good as the TheRnet547v1.

Answer: We modified paragraph 4.2.1 by adding text. Please check Line 421 to 424.

Minor:

5- Line 18: … such as complex…

Answer: corrected:  Line 18

6- Line 57: by maintaining…

Answer: corrected:  Line 57

7- Line 162: these types of structures…

Answer: corrected:  Line 173

8- Line 249: The correct Table number must be 6. Also, the CPU specifications has been repeated in the last row.

Answer: corrected:  Line 260

9- Figure 6(b), 7(h, j, l) and 9 in the y-label: 1- the worst

Answer: corrected:  Fig. 6b in Line 310; Fig. 7 (h,j,i) in Line 348, 350, 352

10- There is a lot of unnecessary and repetitive features in Figure 8: Since the value of each segment in each matrix is given, the colorbars sound unnecessary. Also, the title of subfigures Normalized Confusion Matrix (%) can be removed and mentioned once in the figure caption.

Answer: Fig. 8 modified and added Appendix A. Line 356, 364 to 370.

11- Line 360: …training parameters of…

Answer: corrected:  Line 382

12- In caption of Figure 7, it should be mentioned which subfigures are associated with aug1 and which with aug2. Just mentioning it in body text is not sufficient: the figures must be independently readable.

Answer:  Line 339 to 354.

13- Similar to previous comment, in caption of Figure 8 it should be mentioned what (i), (ii) and (iii) refer to.

Answer:  Line 364 to 369.

14- Similar to previous comment, in Figures 10-15 the subfigures (a), (b), (c) etc. should be introduced and described in the captions.

Answer: Fig. 10 - Line 402 to 407; Fig. 11 – Line 408 to 413; Fig.12- Line 432 to 434; Fig. 13- Line 469 to 480; Fig.14- Line 494 to 498; Fig.15 – 507 to 512.

15- Line 396: compared with.

Answer:  Line 428.

16- Line 470: ground truth

Answer: Line 493, 506.

17- Line 517: open field condition

Answer: 552 to 553.

Round 2

Reviewer 1 Report

This paper can be accepted for publication.